# Combinatorial quantification of distinct neural projections from retrograde tracing

Siva Venkadesh [1,2], Anthony Santarelli[3], Tyler Boesen[3], Hong-Wei Dong [3] & Giorgio A. Ascoli [1,2] ✉

Comprehensive quantification of neuronal architectures underlying anatomical brain connectivity remains challenging. We introduce a method to identify distinct axonal projection patterns from a source to a set of target regions and the count of neurons with each pattern. A source region projecting to $n$ targets could have $2^n-1$ theoretically possible projection types, although only a subset of these types typically exists. By injecting uniquely labeled retrograde tracers in $k$ target regions ($k < n$), one can experimentally count the cells expressing different color combinations in the source region. The neuronal counts for different color combinations from $n$-choose-$k$ experiments provide constraints for a model that is robustly solvable using evolutionary algorithms. Here, we demonstrate this method's reliability for 4 targets using simulated triple injection experiments. Furthermore, we illustrate the experimental application of this framework by quantifying the projections of male mouse primary motor cortex to the primary and secondary somatosensory and motor cortices.

The mouse brain contains over 70 million neurons. To chart how this overwhelmingly large number of neurons are interconnected is a core mission of the BRAIN Initiative to advance our understanding of the structural and functional organizational principles of the mammalian brain[1]. Along with rapid developments of genetic and viral sparse labeling, 3D microscopic imaging, and computational tools for single neuron reconstructions, tens of thousands of single neurons have been reconstructed with detailed axonal trajectories[2–4]. Yet, a comprehensive whole brain wiring diagram at single neuron resolution remains a formidable challenge because of the sheer complexity of the brain and the laborious work of neuronal reconstruction methods[5]. Most anatomical regions in the mammalian brain project to multiple distinct locations[6]. A fundamental relation between macroscopic regional connectivity in the brain and microscopic cellular architecture is that if a source region projects to a target region, there must be at least one neuron type with soma located in the source region whose axon extends to the target region[7]. Consider a source region projecting to $n$ target regions. Several kinds of axonal architecture can possibly

serve as cellular substrates. For instance, there could be $n$ distinct groups of neurons, each extending their axons to a single target region. Alternatively, or in addition, some neurons might sprout their axonal branches into all $n$ target regions. Many additional groups of neurons may exist, each reaching a distinct subset of the $n$ target regions. We maintain that sets of neurons with distinct projection patterns (e.g., a neuron projects to regions a, b, c, d; another neuron projects to regions b, c, and e) belong to different classes. Thus, identifying the projection pattern of each neuron constitutes a form of neuronal classification.

A source region projecting to $n$ target regions may potentially contain up to $2^n - 1$ projections neuron types based on distinct patterns of axonal presence or absence in each target region. If we wish to find the numbers of neurons in each of the potential types (the projectomics census), we need $2^n - 1$ integers. A brute force approach to this challenge might entail labeling individual neurons to visualize their axonal projections[2–4,8,9]. However, this requires a representative sample for each class, which demands an impractical number of

[1]Interdisciplinary Program in Neuroscience, George Mason University, Fairfax, VA 22030, USA. [2]Center for Neural Informatics, Structures, and Plasticity, George Mason University, Fairfax, VA 22030, USA. [3]UCLA Brain Research & Artificial Intelligence Nexus, Department of Neurobiology, David Geffen School of Medicine, University of California Los Angeles, Los Angeles, CA 90089, USA. ✉e-mail: ascoli@gmu.edu

reconstructions while still missing the rarest classes. Moreover, in such an approach, each source region requires its own set of experiments. In this paper, we introduce a practical and scalable strategy to estimate the number of neurons with each distinct projection pattern from multi-label retrograde tract tracing.

## Results

### The concept behind combinatorial projectomics

Suppose we could inject in each of the $n$ target regions a uniquely labeled retrograde tracer, corresponding to colors $c_1, c_2, \ldots c_n$. When analyzing the somata in the source region, those co-labeled with all the $c_1, c_2, \ldots c_n$ colors would correspond to the class of neurons projecting to all $n$ regions. Those co-labeled only with colors $i, j$ and $k$, but none of the other colors, would correspond to the class projecting to regions $i, j$ and $k$ and not to the other regions. Counting the cells with each combination of labels, possibly leveraging recent progress in automation and computer vision[10], would solve the census challenge. Importantly, this analysis can be carried out in parallel on all source regions projecting to the $n$ target regions.

The above-described thought experiment has two major limitations. First, the number of regions targeted by a typical source region in the mammalian brain is larger than the number of distinct retrograde tracers that can be practically injected in vivo. For example, individual source regions in the mouse neocortex project from ~5 to more than 30 potential brain-wide targets[11,12], whereas state of the art tract tracing is limited to triple or at most quadruple injections[13]. Second, retrograde tracer injections must be confined to a portion of the target region in order to minimize the risk of spilling into adjacent regions, which would contaminate the results[14]. Thus, in a subset of neurons of the source region that do project to the given target region, but not to the injected portion, the soma will be unlabeled and thus missed in the cell counts.

Here we introduce an experimental and analytic design that overcomes both of the above limitations. The basic idea is to perform multiple retrograde tracing experiments each covering a subset of the target regions. This is conceptually analogous to the shotgun sequencing strategy in genomics[15]. Every experiment allows the determination of the number of somata in the source region that are co-labeled by any combination of retrograde tracers. It is worth mentioning here that the ordering of the subset of target regions (permutations) selected in an experiment is not considered, and the proposed methodology only considers their distinct combinations. The target regions not covered by a given experiment will contribute certain free variables to the count of neurons with each projection pattern, corresponding to the first limitation. Moreover, the exact proportion of neurons that project to the target regions covered by a given experiment, but not labeled due to the second limitation, will contribute additional free variables. Different experiments cover every target region several times in different combinations, creating a many-to-many relation between the counts of co-labeled somata and the free variables across sets of multiple retrograde tracings. The key to the solution is to obtain several numerical constraints sufficient to estimate all free variables. In other words, enough experiments must be carried out so that the number of co-labeled somatic counts is sufficiently greater than the number of free variables that need to be found.

To explain this approach with an example, consider a source region projecting to four target regions. This scenario yields 15 ($2^4 - 1$) possible projection patterns and corresponding potential neuron types: 4 types projecting to just one of the targets, 6 types projecting to two targets, 4 types projecting to 3 targets, and 1 projecting to all 4 targets (Fig. 1). Now suppose we can only inject three retrograde labels, conveniently referred to as green, red, and blue. We then perform four experiments, each leaving out one of the four target regions. The first experiment injects the green retrograde label in the first target region, the red one in the second, and the blue in the third, leaving out the

fourth target region. From this experiment we can count the number of cells in the source region that are only labeled green, only red, or only blue; those that are co-labeled with each of the three two-color combinations, and those that co-labeled by all three colors, for a total of seven distinct numerical values. Which of the 15 neuron types contribute to the count of the somata that are only labeled green? All of those cells must project to the first target, but not all cells that project to that target will be colored green due to the second limitation. Moreover, the green-only cells also include the neurons projecting to both the first and the fourth target, since the latter was not injected. Lastly, we need to account for the cells projecting to both the first and second target which were not labeled red and similarly for all other neuron types that project to multiple targets as long as they include the first one.

To quantify these contributions, we adopt the following notation: let $G$ be the count of somata exclusively labeled green, $T_1$ the number of neurons that project only to the first target, and $k_1$ the proportion of neurons projecting to the first target that are labeled green (where $k_1 < 1$ due to the second limitation). Similarly, $T_2$ and $k_2$ are the number of neurons that project only to the second target and the proportion of those neurons that are labeled red, respectively (and same for $T_3$, $k_3$ etc.). Furthermore, $T_{12}$ represents the number of neurons that project just to the first and second target and, by extension, $T_{1234}$ is the number of neurons that project to all four targets. We can then formulate the following equation:

$$
\begin{aligned}
G = {} & k_1 \left( T_1 + T_{14} \right) + \\
& k_1 \left( 1 - k_2 \right) \left( T_{12} + T_{124} \right) + \\
& k_1 \left( 1 - k_3 \right) \left( T_{13} + T_{134} \right) + \\
& k_1 \left( 1 - k_2 \right) \left( 1 - k_3 \right) \left( T_{123} + T_{1234} \right)
\end{aligned}
\tag{1}
$$

For each of the four experiments, we can similarly write seven such equations (see Supplementary Equations 1), corresponding to counts for every color combination ($G$, $R$, $B$, $GR$, $GB$, $RB$, $GRB$). In this simple scenario of four targets and three retrograde injections, we thus have 28 equations (constraints) from four experiments and 27 unknowns: 15 numbers of neuron types and 12 retrograde yields (three for each experiment).

It is important to note that the scope of this approach excludes non-projecting types. In other words, this approach aims to quantify the distinct types of projections that actually exist between a source region and a given set of target regions that were included in retrograde injection experiments. However, this approach does not aim to quantify *all* neuron types of the source region, many of which may project to different regions not included in the injection experiments.

In addition, we wish to note that with quadruple retrograde tracing we can in principle run $\binom{n}{4}$ distinct experiments. Each experiment will give us the observed number of cells that project to at least 1 of four classes, 2 of four, 3 of four, or 4 of four (total 15 observations). Every retrograde injection also contributes an additional real value variable, namely the fraction of cells projecting to that region that are in fact labeled. For example, if $n = 7$, we have 127 potential neuron types ($2^7 - 1$) and thus need to estimate 127 counts. There are 35 distinct experiments $\binom{7}{4}$ and each will provide 15 observations and require estimation of 4 additional variables. The total number of equations is thus $35 \times 15 = 525$ and the number of unknowns is $127 + 35 \times 4 = 267$. Note that a $\binom{7}{4}$ model presents a higher constraints-to-unknowns ratio (525:267) than the simpler $\binom{4}{3}$ model described earlier (28:27).

Although only positive values are acceptable for the model solution, the high degrees of non-linearity in the system of equations present multiple positive solutions, when the system is not sufficiently constrained. In the next section, we show that the solutions could become more reliable and robust to experimental error even in the simpler $\binom{4}{3}$ model by repeating an experiment which increases the

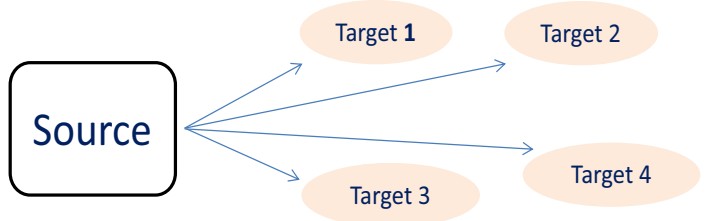

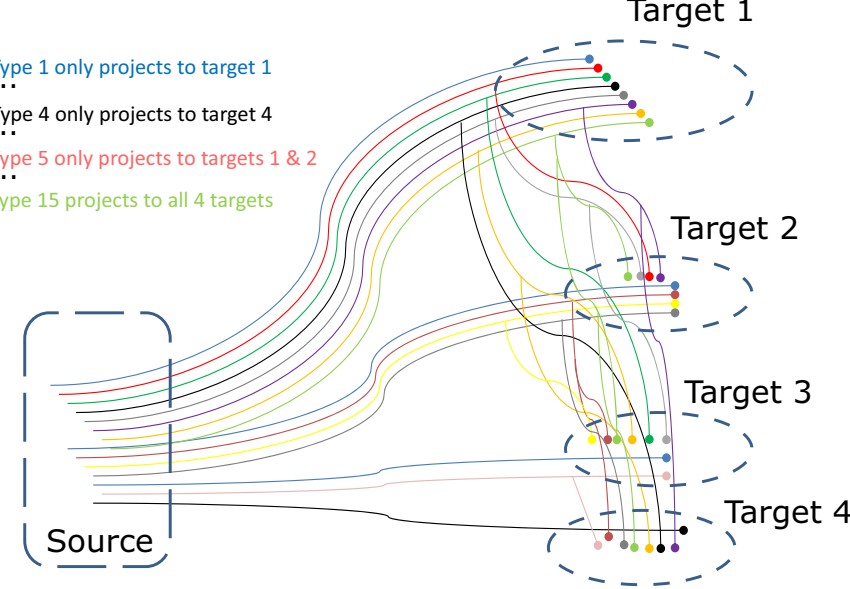

**Fig. 1 | A schema representing potential cellular architectures in the connectivity between a source region and four target regions.** Types of source neurons (S1-4) projecting (>) to one target (T): S1 > T1, S2 > T2, S3 > T3, S4 > T4. Types of source neurons (S5-S10) projecting (>) to two targets (T): S5 > T1,T2; S6 > T1,T3; S7 > T1,T4; S8 > T2,T3; S9 > T2,T4; S10 > T3,T4. Types of source neurons (S11-S13) projecting (>) to three targets (T): S11 > T1,T2,T3; S12 > T1,T3,T4; S13 > T2,T3,T4. Types of source neurons (S14) projecting (>) to four targets (T): S14 > T1,T2,T3,T4.

number of constraints by 7 while only adding 3 unknown variables. This is practically significant, since this allowed us to reliably apply the $\binom{4}{3}$ model to the triple-injection retrograde labeling data acquired from the motor and sensory cortices of the mouse brain, which we present in Section "A proof-of-concept experimental application."

**Reliable estimation of the counts of projection patterns**
In this section, we computationally validate the solvability of the model presented in the previous section by simulating retrograde labeling using surrogate counts for the population sizes of different projection patterns (Fig. 2). We also evaluate the extent to which repeated trials of combinatorial labeling increase the reliability in estimating the surrogate counts in a simulated 4-target and 3-injection configuration.

A "triple-injection" refers to parallel injections in 3 of the 4 targets selected. One repeated trial of an injection $inj_i$ refers to a triple injection repeated for the same three targets in $inj_i$ with an assumed variability for the fractions of axons being labeled. Note that each triple-injection or its repeated trial introduces 3 unknown real values, which correspond to the fractions of axons being labeled, but adds 7 constraints. More generally, each of the $\binom{n}{k}$ distinct combinations of injections introduce $k$ unknown real values and $2^k - 1$ constraints. Therefore, $N$ repetitions of all $\binom{n}{k}$ combinations result in a total of $(N+1) \cdot \binom{n}{k}$ injections, $(N+1) \cdot \binom{n}{k} \cdot k$ unknown real values, and $(N+1) \cdot \binom{n}{k}$ constraints. Also note that the number of unknown integers that need to be estimated remains at $2^n - 1$ for any $N \geq 0$. Thus, as $N$ increases, the ratio of the number of constraints to the number of unknown variables also increases (Fig. 3a).

Surrogate counts for each of the 15 projection patterns and the simulations of triple injection experiments are displayed in Fig. 2. Combinatorial models were described for 4-, 8- and 12- triple injections, and each of the models was solved using an evolutionary algorithm (EA)[16]. We ran multiple trials of the EA with different initial conditions to explore the parameter space and identify all possible solutions to the model (see Section "Methods"). Results from this computational analysis are given in Fig. 3. Convergence patterns of the EA populations showed that the 12-triple injection model created an optimization landscape that is more convex than that of the 4-triple injection model (Fig. 3c), although the 12-triple injection model required more EA generations for the population to converge (Fig. 3b) due to higher number of unknown variables that needed to be estimated. In other words, the 12-triple injection model resulted in a narrower range of solutions, which were also more accurate than the 4-triple injection model (Fig. 3c, d). Furthermore, a systematic reduction of the error in the estimated counts (Fig. 3d) from 4- to 8- and 12- triple injections shows that enhancing the combinatorial model description to include repeated trials of experiments can make our approach extremely reliable. Similar systematic reduction in the error was also observed when these analyses were performed for larger counts of projection patterns (see Supplementary Fig. 1), although 20-triple injections were required to reliably estimate the counts totaling ~100,000. This suggests that the accuracy of estimations depends on the total count of all projection patterns, in addition to the number of injection experiments. It should be noted that the repeated injections must be non-unique in the sense that they must introduce variability in

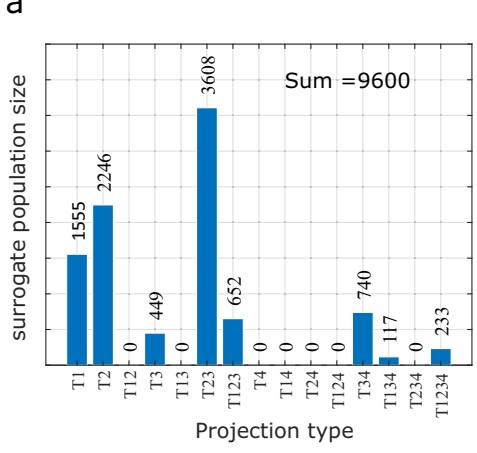

**Fig. 2 | Simulations of triple injection experiments for computational analysis.** **a** Surrogate population sizes for 15 projection patterns. **b** Four distinct combinations of triple injections and repeated trials of each were simulated using values sampled from a normal distribution with mean 0.6 and standard deviation 0.2. These values correspond to the fractions of projecting neurons that are in fact labeled. Two entries in each cell correspond to the counts from two trials of the same experiment for the population sizes given in panel (**a**). Note that the different counts are due to the differences in the fractions of labeled neurons between two trials of the same experiment.

the real values (i.e., fractions of axons labeled), so that they produce distinct constraints. However, such a variability is naturally expected in experimental settings. It is also interesting to observe that the second limitation described in Section "The concept behind combinatorial projectomics" becomes a crucial advantage here.

## A proof-of-concept experimental application

To test our analysis design experimentally, we selected the upper limb area of the primary motor cortex (MOp-ul) of the mouse brain as one source region to quantify its target-specific cortico-cortical projection neurons. Previous work[3,12,17] showed that the MOp contains different subtypes of neurons innervating their targets with a rich variety of collateral projection trajectories. To maximize the eventual yield of successful triple-injections, four retrograde tracers (CTb conjugated with 488, 555, or 647, FG or AAV-retro) were injected in each animal, respectively aiming at four major MOp-ul cortical targets, namely the secondary motor cortex (MOs), the barrel field of the primary somatosensory cortex (SSp-bfd), the secondary somatosensory cortex (SSs), and the rostral MOp (Fig. 4 and Supplementary Fig. 2). Fourteen quadruple injection experiments were completed, with a total of 56 injections. Anatomical locations and sizes of tracer injections were maintained as similar as possible across all experimental cases to maximize the consistency of tracer labeling (for details see Section "Methods"). Post-mortem mapping to the Allen Reference Atlas[18,19] determined the actual locations of the injection sites. This analysis

identified a total of seven sets of triple injections that matched suitable subsets of the four target regions without spillovers. Constraints from these seven successful injection experiments were selected for quantification (see example in Supplementary Fig. 2) in the $\binom{4}{3}$ model. This included 4 distinct triple injections, one repeated trial of one experiment and two repeated trials of another experiment. In addition, three models were described using the segregated counts from MOp-ul layers 2/3, 5, and 6 to delineate layer-specific projection patterns from the source region. Combined and segregated constraints obtained from seven injection experiments are provided in Supplementary Table 1.

The unknown variables of the four models were independently estimated by the solver (see the "Methods" section for computational details). Table 1 provides the estimated counts for each of the 15 projection patterns from the MOp-ul for the combined and the layer-specific models based on the experimental data. Among the four target regions selected, the single projection pattern to SSp was estimated to have the highest population size ($n = 10{,}759 \pm 318$) with the highest contribution from layer 5. The population sizes of neurons projecting to double targets also showed layer-specific differences. For instance, the contribution to the population size of neurons projecting to both MOs and rostral MOp ($n = 623 \pm 310$) was notably less from layer 2/3 compared to layers 5 and 6. Similarly, layer 6 neurons accounted for most of the double-projections to targets SSp and SSs ($n = 1576 \pm 148$), and our method did not find strong evidence of such double-

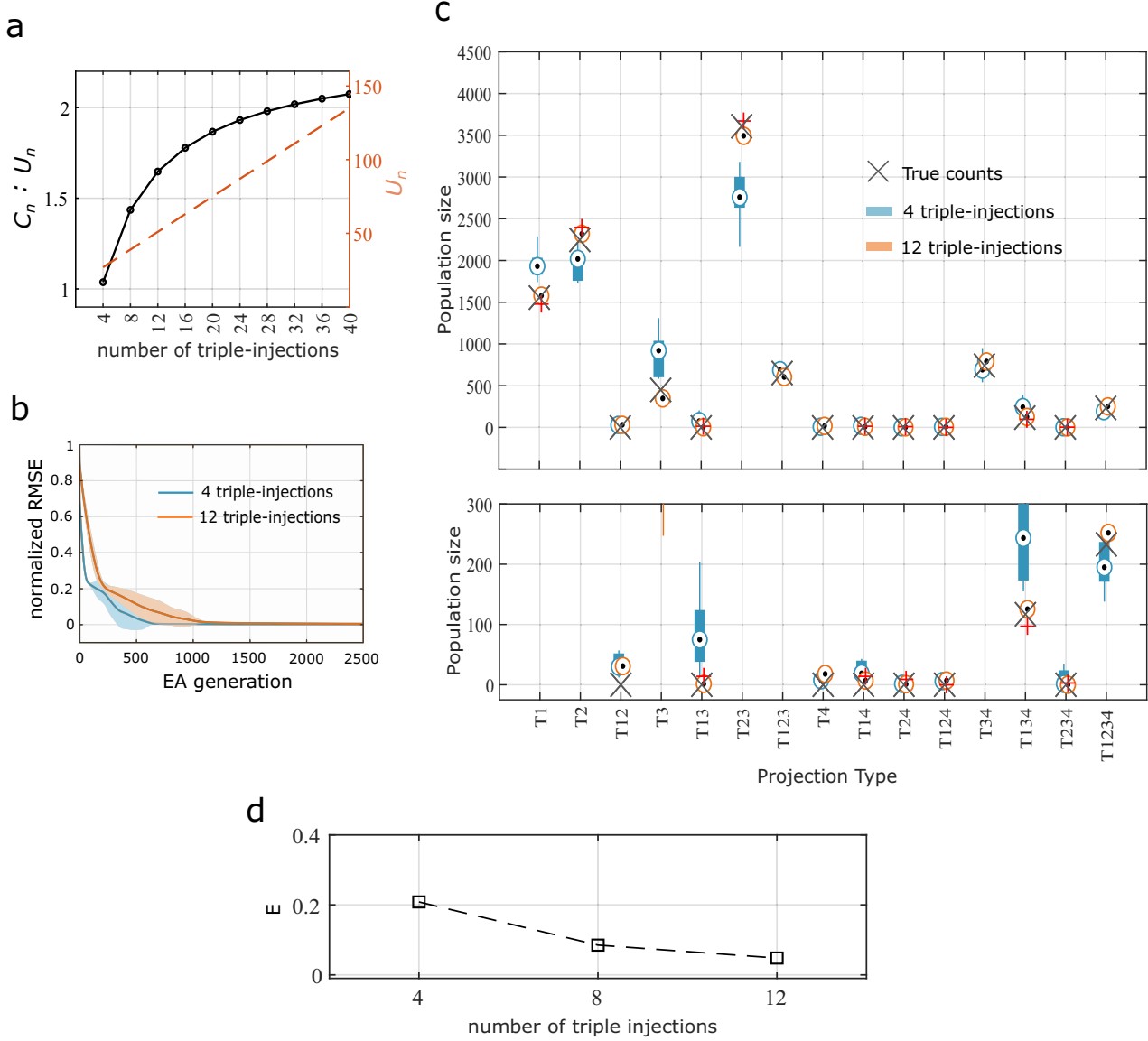

**Fig. 3 | Accurate estimation of surrogate population sizes of projection patterns using an evolutionary algorithm (EA). a** Ratio of the number of constraints ($C_n$) to the number of unknown variables ($U_n$) for increasing number of triple-injections (black). Dotted line represents $U_n$. **b** Evolution of the lowest RMSE on the constraints for 4 distinct combinations of triple-injections (blue) and 2 repeated trials for each totaling 12 injections (orange). RMSE is normalized to the constraint average. Solid lines and shaded areas denote means and standard deviation, respectively. See Fig. 2 for details of simulated triple injections. **c** *Top*: Surrogate population sizes estimated by $N = 10$ independent runs of the EA for each of the two model configurations given in (**b**). Bottom: The range [0, 300] is zoomed in from the top. Central mark indicates the median; the bottom and top edges of the box indicate the 25th and 75th percentiles, respectively; and the whiskers extend to minima and maxima. **d** Average error on the estimated population size (see Section "Methods") for increasing number of triple-injections.

projections from layers 2/3 and 5. Finally, only negligible counts were estimated for the triple and quadruple projection patterns. It is worth mentioning that these results exclude neuron types of the MOp-ul that do not project to any of the regions MOp, MOs, SSp, and SSs.

A post hoc computational analysis was performed using an additional set of surrogate counts that were generated to roughly reflect the distribution of counts estimated (Table 1) from the experimental constraints. The goal was to evaluate the robustness of solution convergence in the region of the search space for seven injection experiments that represented the estimated counts from experimental constraints. Finally, the solver was also run for a null model where the constraints represented random noise, which provided a baseline for comparing the convergence against that of the actual experimental constraints. The solver convergence patterns, and their robustness are given in Fig. 5. While the solutions to the model describing the

surrogate counts converged to lower root-mean square deviation (RMSE) than the model that described actual experimental constraints for seven triple-injections, they both outperformed the null model by a much more substantial margin (Fig. 5).

## Discussion

Connectomics has risen to high prominence in neuroscience in the 18 years since the term was coined[20]. It is now broadly recognized that regional connectivity underlies distributed brain function and single neuron axonal projections underlie regional connectivity[21]. Online 3D atlases of regional connectivity for the mouse brain[11,22] provide an instrumental high-level blueprint of the main communication pathways in mammalian central nervous systems. However, they lack the resolution to identify individual neurons, arguably the key elements for computational function. At the opposite extreme, electron

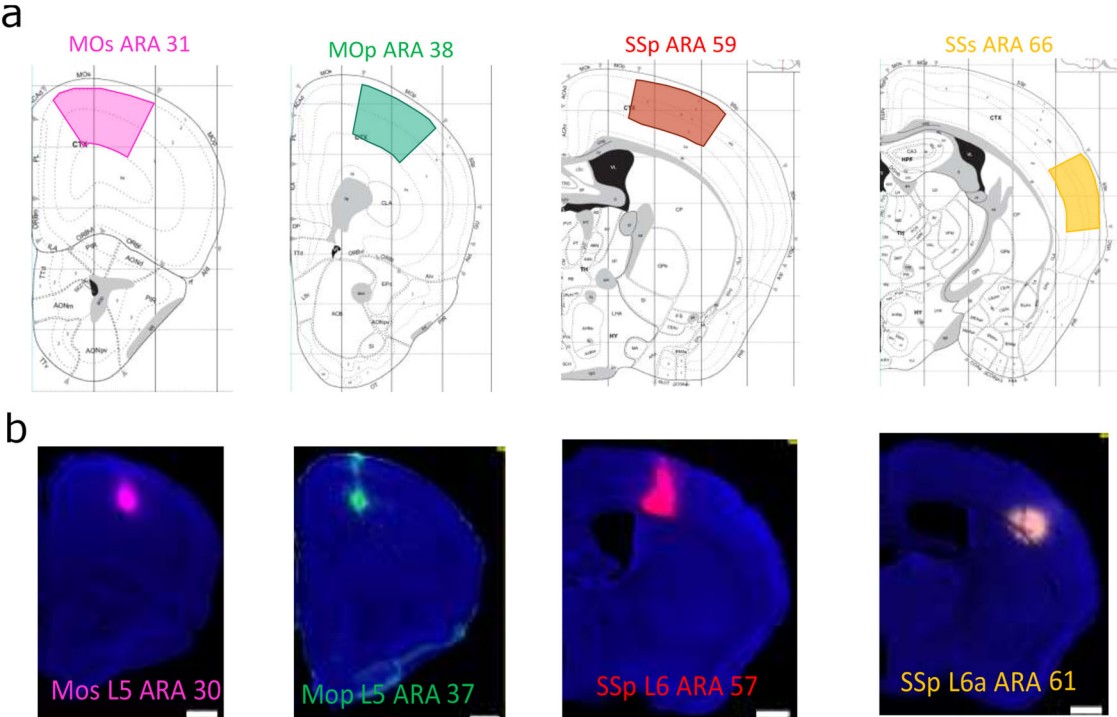

**Fig. 4 | Experimental data obtained from the male mouse primary (p) and secondary (s) somatosensory (SS) and motor (MO) cortices. a** Targets for the retrograde injections described using the Allen Reference Atlas (ARA) areas. **b** The infusion centers from one representative experiment in layers (L) 5 and 6. All scale bars denote 1 mm.

microscopy offers the ultimate opportunity to densely reconstruct every single synapse, but only for local networks of mammalian brains in the foreseeable future[23,24]. Long-range axons constitute the conceptual and physical nexus between brain-wide circuits and synaptic communication. Although single-neuron projection axons can be reliably reconstructed throughout the mouse brain from light microscopy imaging[2,4,8,9], scaling up the digital tracing process remains a formidable open problem[25]. At the same time, the typical divergence of regional connectivity in the mammalian brain poses a combinatorial challenge to the systematic characterization of the neuronal substrates. In this report, we introduced a possible solution based on quantitative analysis of multi-color retrograde injections. Our numerical computations based on realistic surrogate data demonstrated the feasibility, scalability, precision, and robustness of this approach. Moreover, we offered initial empirical evidence of the applicability of the proposed methodology in the case of the mouse primary motor cortex efferent.

The experimental validation of this study is limited by the fact that it included data from only seven injection experiments. Our analysis from Section "Reliable estimation of the counts of projection patterns" (see also Fig. 3) suggested at least 8 triple injection experiments to achieve an average error of roughly 0.1 relative to the true counts. Thus, the high IQRs observed for some of the projection patterns given in Table 1 could be attributed to the sparseness of experimental data included in this study. Furthermore, while it is expected that the estimated total counts for each projection pattern in Table 1 would reflect the sum of their respective layer-specific counts estimated independently, there were differences between the combined and the sum of layer-segregated estimations. In addition to the sparsity of the experimental data, the noise introduced in segregating the layer-specific counts (see Supplementary Table 1) likely enhanced such differences. Therefore, the counts for projection patterns with high IQRs should be interpreted cautiously, and future studies with more injection experiments are required to fully validate the results presented in

Table 1. However, our analyses of surrogate and real data collectively show that the population sizes of various projection patterns between a source and four target regions can be reliably estimated given sufficient experimental constraints using the model presented in this study.

Our results are in fact consistent with those reported in the literature. Neuronal connectivity of the MOp has been studied extensively at macro- (regional specific)[3,12,26–28], meso- (cell type-specific)[3,22,29] and micro-scales (single neuron)[2–4]. At macroscale (regional specific), the MOp-ul shares extensive reciprocal connections with multiple domains of the MOs, SSp and SSs[12]. At the meso-scale (cell type specific), cortico-cortical connections arise mostly from the intratelencephalic (IT) neurons across layer 2–6; while other two major classes of neuron types, pyramidal (PT) and cortico-thalamic (CT), generate much less collateral projections to other cortical areas[3,17,26–29]. In a recent study combining viral sparse labeling, 3D microscopic imaging, and computational algorithms, detailed axonal projection trajectories of ~300 individual neurons in the MOp were reconstructed[2–4], in principle providing the initial core of a ground truth dataset for validating all possible axonal patterns shown in the current study (Table 1). However, that cell type-specific single neuron reconstruction strategy relied on available Cre-driver mouse lines, as well as labor intensive and time-consuming 3D imaging and computational reconstruction procedures. Consequently, only a small fraction of MOp neurons was reconstructed, providing insufficient information to generate a comprehensive landscape of individual neuronal projection motifs. This inadequacy highlights the difficulty of obtaining sufficiently large sample sizes even with big data consortium efforts, underscoring the need for practical and scalable alternatives. In this perspective, our current combinatorial approach provides an important complementary approach for cataloging connectivity-based neuronal types in the mammalian brain—one major goal of contemporary neuroscience research.

**Table 1 | Estimated population sizes of projection patterns between MOp-ul and the target regions based on the experimental data**

| Projection Pattern | MO_p (combined) | | Layer 2/3 | | Layer 5 | | Layer 6 | |
|---|---|---|---|---|---|---|---|---|
| | Median | IQR[c] | Median | IQR | Median | IQR | Median | IQR |
| $T_1$ [a] | 1531 | 425 | 405 | 78 | 623 | 1119 | 725 | 139 |
| $T_2$ | 3291 | 218 | 2333 | 1658 | 92 | 584 | 840 | 152 |
| $T_{12}$ | 623 | 310 | 40 | 8 | 549 | 352 | 156 | 26 |
| $T_3$ | 10,759 | 318 | 4558 | 2057 | 6997 | 904 | 1649 | 607 |
| $T_{13}$ | 683 | 170 | 65 | 23 | 449 | 583 | 628 | 569 |
| $T_{23}$ | 1267 | 117 | 383 | 211 | 700 | 213 | 561 | 56 |
| $T_{123}$ | 7 | 4 | 47 | 39 | 169 | 153 | 4 | 2 |
| $T_4$ | 0 | 1 | 27 | 32 | 40 | 76 | 1 | 1 |
| $T_{14}$ | 0 | 1 | 10 | 11 | 2 | 2 | 1 | 1 |
| $T_{24}$ | 70 | 116 | 8 | 7 | 112 | 60 | 9 | 12 |
| $T_{124}$ | 84 | 134 | 0 | 4 | 2 | 38 | 3 | 3 |
| $T_{34}$ | 1576 | 148 | 0 | 2 | 2 | 13 | 1047 | 631 |
| $T_{134}$ | 1 | 120 | 0 | 0 | 0 | 1 | 141 | 638 |
| $T_{234}$ | 1 | 4 | 0 | 0 | 0 | 2 | 122 | 53 |
| $T_{1234}$ | 2 | 3 | 0 | 0 | 0 | 0 | 4 | 5 |
| SUM | 20,099[b] | 405 | 9047 | 2372 | 10827 | 2182 | 6168 | 177 |

[a]Subscripts of projection patterns denote the target (T) areas MOp (1), MOs (2), SSp (3), and SSs (4).
[b]Median of the sum.
[c]Interquartile range.

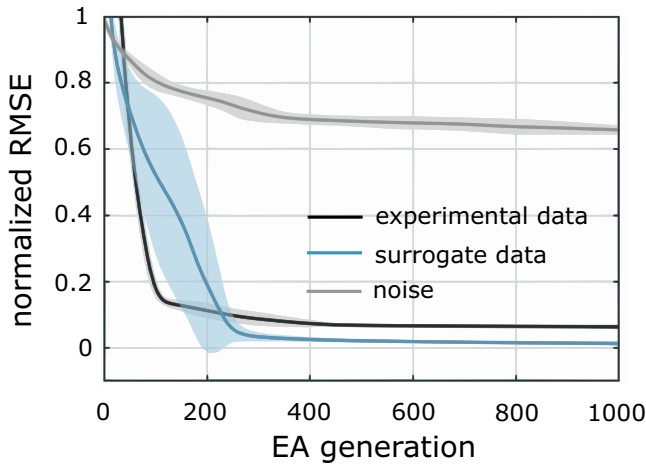

**Fig. 5 | EA convergence patterns for experimental, surrogate, and random constraints.** Evolution of the lowest RMSE for the counts obtained from the injection experiments performed in MOs, MOp, SSp, and SSs (black), surrogate counts generated to reflect the distribution of estimations given in Table 1 (blue), and randomly generated counts for various color combinations in a simulated source (gray). Solid lines and shaded areas denote means and standard deviation, respectively.

Another recent development with great potential for comprehensive high-throughput mapping of cell type-specific axonal projections is MAPseq[30]. This technique, based on anterograde bar-coding, was successfully applied to the mouse primary motor cortex in a recent consortium investigation involving over a dozen independent laboratories[3]. However, the specific MOp area injected in that study cannot be reliably deemed to significantly overlap with the locations of the multi-labeled cell bodies obtained in the experiments described in the present report. Furthermore, MAPseq data collection relies on previous knowledge or anterograde tracing data as guidance to dissect the targeted brain regions for acquiring transported barcoding molecules. Thus, the granularity and accuracy of MAPseq-based axonal

projection information heavily depends on laser-capture tissue microdissection sampling densities and anatomical accuracy of each targeted area. Because those operations are conducted in fresh frozen, thick sections without histological staining (e.g., Nissl), it remains very challenging to dissect tissues accurately following their anatomical borders, especially for smaller brain structures as in the present study. The above limitations prevent a direct comparison of our current experimental findings with previous MAPseq data.

Notably, the technique described in this report can be applied to increasingly large amounts of retrograde tract tracing data that are being systematically collected and deposited in the BICCN data portal and other open resources[1,7,21]. Simulations from the current study show that the model becomes highly reliable for the $\binom{4}{3}$ configuration (Fig. 2c, d) even with 12 triple injections. Alternatively, a $\binom{7}{4}$ model, which has a high constraints-to-unknowns ratio, could potentially quantify projection patterns for 7 regions (127 types) without repeated injection experiments. However, a comprehensive application of a $\binom{7}{4}$ model requires 35 distinct quadruple injections and a solver capable of estimating 267 unknown variables in a system where 140 (35 × 4) real variables interact in a non-linear manner. While it is beyond the scope of this study to evaluate the solvability of such a $\binom{7}{4}$ model, our analysis with the simpler $\binom{4}{3}$ model suggests that repeating the injection experiments can in principle increase the reliability of a $\binom{M}{4}$ model for any $M > 4$. The only challenges to this scalability are the cost associated with quadruple injection experiments and the computational cost of solving the (non-linear) models with many unknown parameters. Nevertheless, we have shown in this paper that our approach can, in principle, reliably and robustly quantify the cellular architectures of mammalian brain connectivity in a comprehensive manner.

Indeed, retrograde tract tracing methods have been broadly used in the neuroscience field for more nearly 4 decades[31], and one of our labs is one of the pioneer groups to systematically apply multiple fluorescent retrograde tracers in constructing the mouse connectome[3,12,32]. Those previous publications demonstrated the feasibility, robustness, accuracy, precision, and scalability of this technology. Specifically, over the course of multiple studies, we have conducted triple and quadruple fluorescent retrograde tract tracing experiments in over ~600 mice over many brain structures. A large

collection of these data is already publicly available[33]. Based on our own past and ongoing experimental work, the approach described in the present paper can be scaled up to reliably perform 500 quadruple injection experiments, assuming the resources of a large effort such as those presently pursued by the Allen Institute for Brain Science, Howard Hughes Medical Institute Janelia Research Campus, and the National Institutes of Health's BRAIN Initiative Cell Atlas Network (BICAN). Such a dataset could be analyzed with our combinatorial projectomics framework to resolve up to 12 distinct targets, potentially quantifying up to >4000 ($2^{12}$) different projection types at once in each of every source region projecting to those targets.

Differences in intrinsic electrophysiology and input-output features[34], as well as in single cell transcriptomics[35] add to the complexity of neuronal classifications and especially deserve scrutiny when attempting to identify conserved patterns across cortical areas and species. How to effectively combine these distinct dimensions together with the axonal projection patterns of long-range neurons remains an open challenge.

## Methods
### Solving the systems of equations
The systems of equations were solved using evolutionary algorithms (EA). We employed $(\mu + \lambda)$ evolution strategies[16] without adaptive mutation to estimate the integer and the real-valued unknown variables in the combinatorial models. Briefly, $\lambda$ offspring solutions are created from $\mu$ parents, and selection pressure is applied to both parents and offspring solutions for survival into the next generation. A total of 50 trials of EAs were run, each with different initial conditions. The population sizes were set to $\mu = 50,000$ and $\lambda = 250,000$ for all EA runs. An integer random-walk and a gaussian step mutations with mutation rates of 0.1 were applied for the integer and real-valued variables respectively. The total number of EA generations were set to 2500 for all analyses except for the model describing a total surrogate count of ~100,000 (see Supplementary Fig. 1), which used 10,000 EA generations. A Java-based evolutionary computing library (ECJ)[36] was utilized in this study. The ECJ configuration and the full set of EA parameters are described in the shared software[37] under `EqnSolver/input/.params`.

For the simulated experiments, surrogate counts for projection patterns were generated by first setting the counts of seven randomly selected types to zero (since not all possible projection patterns are expected to be present between a source and a set of target regions) and then randomly generating counts for the remaining types. Note that these surrogate counts represented the true counts of projection patterns in the source region that needed to be estimated by the EA. The values corresponding to the fractions of neurons being labeled in each simulated injection experiment were also randomly generated. The counts of neurons in the source with different color combinations following the injection experiments provided constraints for the model. Then, the EAs were run to estimate the counts of all projection types and the fractions of labeled neurons in each experiment by minimizing The Root Mean Square Error (RMSE) on the constraints.

The Medians from the top 10 EA runs with the lowest RMSE represented the adopted solution, and the error in the estimated counts ($E$) in simulated experiments was defined as follows:

$$E = \frac{\sum_{i=1}^{I}\left(1 + \frac{\left(\frac{M}{I} - m_i^t\right)}{M}\right) \times |m_i^t - m_i^e|}{M} \tag{2}$$

$M$ and $I$ are the true sum and number of types respectively, and $m_i^t$ and $m_i^e$ are the true (surrogate) and estimated counts respectively of type $i$. In addition to the medians, interquartile ranges (IQR) were calculated from the top 10 EA solutions for the analysis involving experimental data. The IQR characterized the spread of estimated solutions in the parameter space for each projection pattern.

## Mouse Connectome Project methodology: multiple fluorescent retrograde tracing strategy
Anatomical tract tracing data was generated as part of the Mouse Connectome Project (MCP) following experimental methods and online publication procedures as described previously[12,38–40]. To retrogradely label projection neurons in the upper limb of the mouse primary motor cortex (MOp-ul), we used a multiple tracing method to simultaneously inject different fluorophore-conjugated retrograde tracers into different neocortical projection targets (up to 4) of the MOp. These injection sites were pre-selected based on anterograde tracing results with injections into the MOp-ul as shown in previous publications[3,12].

**Animal subjects.** All MCP tract-tracing experiments were performed using 8-week-old male C57BL/6J mice (Jackson Laboratories). Mice had ad libitum access to food and water and were group-housed within a temperature- (21–22 °C), humidity- (51%), and light- (12 h light/dark cycle) controlled room within the Zilkha Neurogenetic Institute vivarium. All experiments were performed according to the regulatory standards set by the National Institutes of Health Guide for the Care and Use of Laboratory Animals and by the institutional guidelines described by the University of Southern California Institutional Animal Care and Use Committee.

**Tracer injection experiments.** The MCP uses a variety of combinations of anterograde and retrograde tracers to simultaneously visualize multiple anatomical pathways within the same Nissl-stained mouse brain. Retrograde tracers included cholera toxin subunit B conjugates 647, 555 and 488 (CTb; AlexaFluor conjugates, 0.25%; Invitrogen), Fluorogold (FG; 1%; Fluorochrome, LLC), as well as AAVretro-cre. Quadruple retrograde tracer experiments involved four different injections sites receiving a unique injection of either 0.25% CTb-647, CTb-555, CTb-488, and 1% FG. Because each retrograde tracer has its own specific properties, including but not limited to speed of migration, uptake into fibers of passage, extent of diffusion, we randomized the assignment of each tracer to a distinct target region. Thus, the axonal projections to each target region are captured through diverse tracers across multiple experiments and, conversely, each tracer is used in all target regions over the course of the whole study. This approach minimizes tracer-specific experimental bias.

**Stereotaxic surgeries and histology and immunohistochemical processing.** On the day of the experiment, mice were deeply anesthetized and mounted into a Kopf stereotaxic apparatus where they were maintained under isofluorane gas anesthesia (Datex-Ohmeda vaporizer). For quadruple retrograde tracing experiments, 50 nl of retrograde tracers were individually pressure-injected via glass micropipettes at a rate of 10 nl/min (Drummond Nanoject III). All injections were placed in the right hemisphere.

After 4–6 days post-surgery, each mouse was deeply anesthetized with an overdose of Euthasol (pentobarbital) and trans-cardially perfused with 50 ml of 0.9% saline solution followed by 50 ml of 4% paraformaldehyde (PFA, pH 9.5). Following extraction, brain tissue was post-fixed in 4% PFA for 24–48 h at 4 °C. Fixed brains were embedded in 3% Type I-B agarose (Sigma-Aldrich) and sliced into four series of 50μm thick coronal sections using a Compresstome (VF-700, Precisionary Instruments, Greenville, NC) and stored in cryopreservant at −20 °C. All sections were stained with Neurotrace 435/455 (Thermo Fisher Cat# N21479) for 2–3 h to visualize cytoarchitecture. After that, sections were mounted onto glass slides and cover slipped using 65% glycerol.

**Imaging and post-acquisition processing.** The tissue sections were scanned on an Olympus VS120 slide scanning microscope with 10× objective. Each tracer was visualized using appropriate fluorescent

filters and whole tissue section images were stitched from tiled scanning into VSI image files. Raw images were corrected for left-right orientation and matched to the nearest Allen Reference Atlas[18,19] coronal levels. An informatics workflow was specifically designed to reliably warp, reconstruct, annotate, and analyze the labeled pathways in a high-throughput fashion through our in-house image processing software Connection Lens[38,39]. Threshold parameters were individually adjusted for each case and tracer, resulting in binary image output files suitable for quantitative analysis. Adobe Photoshop was used to correct conspicuous artifacts in the threshold output files that would have spuriously affected the analysis. A separate copy of the atlas-registered TIFF image file was brightness/contrast adjusted to maximize labeling visibility and images were then converted to JPEG file format for online publication in the MCP iConnectome viewer (MouseConnectome.org).

**Assessment of injection sites.** All injection cases included in this work are, in our judgment, prototypical representatives of each brain area. We have previously demonstrated our targeting accuracy with respect to injection placement, our attention to injection location, and the fidelity of labeling patterns derived from injections to the same location (see Supplementary Methods in our previous reports[12,39] for details).

**Data analysis.** For each brain, eight serial sections covering ARA 45–59 of the MOp were used for quantification. Sections were cut at 50 μm; 200 μm were present between each serial section. Retrogradely labeled neurons were revealed respectively by fluorescence of CTB conjugated with Alexa Fluor 488, 555, 647, and FG (in some cases, AAVretro-Cre also was used). NeuroTrace 435/455 (Blue fluorescent Nissl stain; Invitrogen) revealed cytoarchitectonic background of each section to ensure accuracy of anatomical identification of those retrogradely labeled neurons. Sections were scanned on the Olympus VS –120 virtual slide microscope. Individual channels were exported to Photoshop and overlaid for manual annotation. Cells containing positive signal in each channel were annotated with a 10-pixel point. Distinct annotations with overlap > 80% were noted to be positive for each annotated tracer, and a combination point was made. Annotations were quantified using ImageJ. To avoid over-counting, each annotated cell was recorded only once in the datasheet, with a 3x tracer positive cell being absent from the six contributing 2x combinations and three contributing single tracer positives. MOp layers were then parcellated into layers 1, 2/3, 5, and 6 based on their cytoarchitectural properties. Quantification of the annotated cells were then ascribed to each layer.

### Reporting summary
Further information on research design is available in the Nature Portfolio Reporting Summary linked to this article.

## Data availability
All the imaging data underlying the results described in this work are publicly accessible without restrictions or credentials at https://doi.org/10.5281/zenodo.10048805.

## Code availability
The software to reproduce the results included in this paper is available at https://doi.org/10.5281/zenodo.8416755[37].

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

## Acknowledgements

The authors thank Drs. Rebecca Goldin, John Darrell Van Horn, Saleet Jafri and Donna Fox. This work was supported by the NIH grants R01NS39600, U01MH114829, and RF1MH128693. S.V. was supported by the first two of those grants until 5/29/2020 in part through this project from 11/27/2017.

## Author contributions

G.A.A. and H.W.D. conceived the project. S.V. contributed to model enhancement and numerical analyses. A.S., T.B., and H.W.D. acquired experimental data. S.V., G.A.A., A.S., and H.W.D. contributed to manuscript preparation. S.V., G.A.A., and H.W.D. contributed to manuscript revisions.

## Competing interests

The authors declare no competing interests.
