## [Peer Review File · Nature Communications]

Combinatorial quantification of distinct neural projections from retrograde tracingREVIEWER COMMENTS

Reviewer #1 (Remarks to the Author):

Dr. Ascoli and his coworkers present a solution for neuronal classifications based on a combinatorial quantification of distinct, repeated, multi-label retrograde tract tracing experiments. The authors run simulated experiments with n targets applying complicated mathematics, and include a proof of concept model experiment, that in a source (Mop) and 4 targets (MOs, SS—bfd, SSs and rostral Mop), have only 15 cellular types, based on various degree of colocalization of axons to these 4 targets.

This reviewer acknowledges with some sad resigns that the trend in the current 'industrial' neuroscience shifted to quantify everything and left for future times analysis tons of data and an attempt understanding the significance of the ever-increasing details of subtle circuits. On the other hand, some of the AI champions suggest that we do not need a detailed anatomy and the 3D blueprints can easily be smashed since AI will solve the organization of every function. Anyway, using even clever combination of retrograde tracers, problems remain that each retrograde tracer has its own specifics in terms of speed of migration, uptake into fibers of passage, extent of diffusion, etc., not to mention differences of intrinsic electrophysiology, lack of input-output features and single cell transcriptomics that obviously will add to the complexity of cellular classifications before reasonable conclusion can be reached about cell 'clusters' that are conserved across cortical areas and species and indeed deserve more scrutiny.

Reviewer #2 (Remarks to the Author):

The paper "Combinatorial quantification of distinct neural projections from retrograde tracing" describes a method for quantifying the different axonal projection patterns from a source region to a set of target regions in the brain. This method involves injecting uniquely labeled retrograde tracers in a subset of the target regions and counting the cells expressing different combinations of colors in the source region. By performing experiments for a subset of different combinations of target regions, the counts of cells with different targeting patterns provide constraints for a system of equations that include unknown variables corresponding to the count of neurons for each projection pattern.

The authors propose a brilliant mathematical approach as a solution to a hard experimental problem. I express my admiration for this endeavor, and I hold the belief that this methodology will be of immeasurable value to the scientific community. The authors have effectively addressed the limitations of the study and presented compelling evidence in support of the proposed methodology, both in theoretical and experimental contexts.

Below are some suggestions to enhance the clarity and conciseness of the text in order to effectively communicate the important findings of the study to a larger audience.

Major suggestions

1. Missing use case "zero projection"

I have observed that while the authors correctly calculate all possible combinations, they have not accounted for the scenario in which cells do not project to any of the chosen target regions. It would be advisable for the authors to include this use case since some cells in a region may not project to any of the target regions under examination, and this should heavily impact the results of the analytical computations. In the event that this is not feasible biologically, the authors should provide a comprehensive explanation and justification, which I did not find in the text.

2. Visualizations of the methods and results

In order to enhance the clarity of the methodology and facilitate reader comprehension, the visualizations of both the methods and results should be improved significantly. I provide a few examples of potential improvements, but the authors have the discretion to determine the specific visualizations that are most appropriate. It is my hope that these suggestions may inspire enhancements in this area.

Figure #2:

The injections and the resulting targets are visualized with the same three colors but it is my impression that color "blue" for instance corresponds to different targets at each experiment. This is confusing as it is not clear which region is studied in each injection experiment, and the results for the same colors do not correspond to the same injected regions. For example 1944 blue in experiment one and 502 blue in experiment two do not correspond to the same region. It might be beneficial to illustrate the results for the combinations like in panel Fig2a, to have a more clear representation of the findings.

The visualization of combinations presented in panel 2b can be challenging to comprehend. A more effective approach would be to use an alternative representation for the combinations. For example it would be more effective to create an adjacency matrix to present the results (see attached pdf for detailed sketches).

It should be noted that the representation provided is just one of many alternatives that can effectively demonstrate the same effect.

Minor suggestions

1. Distinction between combinations versus permutations

While both refer to grouping elements of a set into subsets, in a permutation the elements of a subset are ordered, while in a combination that is not the case (all orderings are equivalent). I believe that the proposed methodology focuses on the latter and I think it would be important to clarify this distinction: the ordering of parsing of regions is not taken into account and only the presence or absence of connections between source and target regions can be investigated. This presents an opportunity for future directions to enhance the methodology, with the goal of determining the ordering of targets.

2. Simple validation of the methodology

Since the authors have thoroughly investigated the used case of four projection targets by collecting experimental data, there is a simple validation that can be performed to demonstrate the validity of the methodology. For a subset of region targets (3) they have performed the experiments to label all three targets and computed the cell counts for each of the 7 ($2^3 - 1$) possible target combinations. Provided that sufficient experiments are available for each combination, the authors could perform the same analysis that they did for ($n=4$, $k=3$) to the ($n=3$, $k=2$) use case in which the solution could be inferred from the existing experiments. The solution for ($n=3$, $k=2$) can be analytically computed and compared to the experimental results, thus providing a trivial validation of the method and strengthening the authors' arguments.

3. Distinction between experimental and theoretical results

To prevent confusion between experimental and computational results, it would be beneficial for the reader if the authors clearly distinguished between the two in both the text and figures. This will enable the reader to discern the type of results being presented with ease.

I would like to commend the authors for their meticulous work and dedication to this manuscript, and express my best wishes for their continued progress in this direction of valuable research.

Lida Kanari

Reviewer #3 (Remarks to the Author):

The authors provide a compelling analytical framework to determine the distinct projection profiles of single neurons from one source region to a set of target regions. This is potentially a very powerful and useful development. I do not have the background to determine the mathematical validity of the approach, but it appears well reasoned, developed and I see no obvious flaws. However, I have two major concerns about the work regarding its suitability to the wide-readership of Nature Communications.

First, the paper does not provide a ground truth or orthogonal data set to demonstrate that a combination of practical, even if difficult, experiments and the proposed analytical approach will capture the different projection patterns of individual neurons. Both single cell reconstructions and MAPseq could potentially provide these datasets.

Second, the authors acknowledge that producing data sets with more than 3-4 colors is about the limit of what is possible. Given that many brain areas, including the one investigated usually innervate at least 10, and can innervate more than 20 targets, it remains unclear how widely applicable or scalable this approach currently is, or will be in the future.

I believe the authors have demonstrated the feasibility and robustness of their approach, but I do not believe they have demonstrated how it scales or its accuracy/precision. The scaling problem is mainly experimental that can either involve being able to perform both many accurate injections and subsequently detect and separate the many colors in the source brain region; or performing many sets of 3-4 color experiments to reveal the full set of projection patterns. To demonstrate the accuracy and precision of the method requires other data sets, e.g. single cell reconstructions and/or barcoding methods such as MAPseq. The authors eloquently address the difficulty of this issue in the discussion, highlighting the need for new approaches, However, I do feel it needs to be addressed more directly and thoroughly than is presented in the manuscript.

This work is clearly of high quality but as currently presented appears more suited to a more specialized journal.

Review (NCOMMS-23-04112-T)

The paper “*Combinatorial quantification of distinct neural projections from retrograde tracing*” describes a method for quantifying the different axonal projection patterns from a source region to a set of target regions in the brain. This method involves injecting uniquely labeled retrograde tracers in a subset of the target regions and counting the cells expressing different combinations of colors in the source region. By performing experiments for a subset of different combinations of target regions, the counts of cells with different targeting patterns provide constraints for a system of equations that include unknown variables corresponding to the count of neurons for each projection pattern.

The authors propose a brilliant mathematical approach as a solution to a hard experimental problem. I express my admiration for this endeavor, and I hold the belief that this methodology will be of immeasurable value to the scientific community. The authors have effectively addressed the limitations of the study and presented compelling evidence in support of the proposed methodology, both in theoretical and experimental contexts.

Below are some suggestions to enhance the clarity and conciseness of the text in order to effectively communicate the important findings of the study to a larger audience.

Major suggestions

1. Missing use case “zero projection”

I have observed that while the authors correctly calculate all possible combinations, they have not accounted for the scenario in which cells do not project to any of the chosen target regions. It would be advisable for the authors to include this use case since some cells in a region may not project to any of the target regions under examination, and this should heavily impact the results of the analytical computations. In the event that this is not feasible biologically, the authors should provide a comprehensive explanation and justification, which I did not find in the text.

2. Visualizations of the methods and results

In order to enhance the clarity of the methodology and facilitate reader comprehension, the visualizations of both the methods and results should be improved significantly. I provide a few examples of potential improvements, but the authors have the discretion to determine the specific visualizations that are most appropriate. It is my hope that these suggestions may inspire enhancements in this area.

Figure #2:

The injections and the resulting targets are visualized with the same three colors but it is my impression that color “blue” for instance corresponds to different targets at each experiment. This is confusing as it is not clear which region is studied in each injection experiment, and the results for the same colors do not correspond to the same injected regions. For example 1944 blue in experiment one and 502 blue in experiment two do not correspond to the same region. It might be beneficial to illustrate the results for the combinations like in panel Fig2a, to have a more clear representation of the findings.

The visualization of combinations presented in panel 2b can be challenging to comprehend. A more effective approach would be to use an alternative representation for the combinations.

For example it would be more effective to create an adjacency matrix to present the results, such as illustrated below:

Alternatively, I demonstrate below a more elegant mapping using concepts from algebraic topology that could be used as a representation: point-0 level connection, line-1 level connection, triangle-2 level connection, circle-3 level connection. Here are some examples of how this can be used in practice:

It should be noted that the representation provided is just one of many alternatives that can effectively demonstrate the same effect.

Minor suggestions

1. Distinction between combinations versus permutations

While both refer to grouping elements of a set into subsets, in a permutation the elements of a subset are ordered, while in a combination that is not the case (all orderings are equivalent). I believe that the proposed methodology focuses on the latter and I think it would be important to clarify this distinction: the ordering of parsing of regions is not taken into account and only the presence or absence of connections between source and target regions can be investigated. This presents an opportunity for future directions to enhance the methodology, with the goal of determining the ordering of targets.

2. Simple validation of the methodology

Since the authors have thoroughly investigated the used case of four projection targets by collecting experimental data, there is a simple validation that can be performed to demonstrate the validity of the methodology. For a subset of region targets (3) they have performed the experiments to label all three targets and computed the cell counts for each of the 7 ($2^3 - 1$) possible target combinations. Provided that sufficient experiments are available for each combination, the authors could perform the same analysis that they did for (n=4, k=3) to the (n=3, k=2) use case in which the solution could be inferred from the existing experiments. The solution for (n=3, k=2) can be analytically computed and compared to the experimental results, thus providing a trivial validation of the method and strengthening the authors' arguments.

3. Distinction between experimental and theoretical results

To prevent confusion between experimental and computational results, it would be beneficial for the reader if the authors clearly distinguished between the two in both the text and figures. This will enable the reader to discern the type of results being presented with ease.

I would like to commend the authors for their meticulous work and dedication to this manuscript, and express my best wishes for their continued progress in this direction of valuable research.

Lida Kanari

Reviewer #1 (Remarks to the Author):

Dr. Ascoli and his coworkers present a solution for neuronal classifications based on a combinatorial quantification of distinct, repeated, multi-label retrograde tract tracing experiments. The authors run simulated experiments with n targets applying complicated mathematics, and include a proof of concept model experiment, that in a source (Mop) and 4 targets (MOs, SS—bfd, SSs and rostral Mop), have only 15 cellular types, based on various degree of colocalization of axons to these 4 targets.

Thank you very much for the in-depth reading and accurate summary of our manuscript.

This reviewer acknowledges with some sad resigns that the trend in the current ‘industrial’ neuroscience shifted to quantify everything and left for future times analysis tons of data and an attempt understanding the significance of the ever-increasing details of subtle circuits. On the other hand, some of the AI champions suggest that we do not need a detailed anatomy and the 3D blueprints can easily be smashed since AI will solve the organization of every function.

The debate of whether we need detailed anatomy or whether the 3D blueprints can easily be smashed certainly deserves attention, but is possibly better tackled in a separate, dedicated paper. Even if or when AI should solve the organization of every function, some numerical and experimental quantification at this level may be useful or necessary to validate theoretically evolved neural network architectures. We would be willing to include a brief discussion of this possibility in this manuscript, if the Editor believes it could be of interest for the Nature Communication readers. It may be useful to note that this manuscript is under consideration as part of a special package invited by the Nature editors specifically to cover recent anatomy developments from the BRAIN Initiative Cell Census Network (BICCN).

Anyway, using even clever combination of retrograde tracers, problems remain that each retrograde tracer has its own specifics in terms of speed of migration, uptake into fibers of passage, extent of diffusion, etc., not to mention differences of intrinsic electrophysiology, lack of input-output features and single cell transcriptomics that obviously will add to the complexity of cellular classifications before reasonable conclusion can be reached about cell ‘clusters’ that are conserved across cortical areas and species and indeed deserve more scrutiny.

Thank you for bringing up these important points. It is true that distinct retrograde tracers vary in speed of migration, uptake into fibers of passage, and extent of diffusion. This is why we do not always inject the same tracer in the same target region, but instead shuffle the injections among individual experiments. In this way, the axonal projections to each target region are captured through several distinct tracers when analyzing the collection of data (and conversely, each tracer is used in all target regions across experiments). This approach minimizes experimental bias. We added an explanation of this design choice in the Materials and Methods (lines 373-378).

We agree that differences of intrinsic electrophysiology, input-output features, and single cell transcriptomics add to the complexity of cellular classifications and indeed deserve scrutiny, especially when attempting to identify conserved patterns across cortical areas and species. We have now included these considerations in the Discussion (lines 321-325). We also note in the Discussion that axonal projections define the circuit blueprint of the nervous system (lines 229-235), hence new quantitative approaches for their quantification, as presented in this work, can facilitate future scientific breakthroughs.

Reviewer #2 (Remarks to the Author):

The paper “Combinatorial quantification of distinct neural projections from retrograde tracing” describes a method for quantifying the different axonal projection patterns from a source region to a set of target regions in the brain. This method involves injecting uniquely labeled retrograde tracers in a subset of the target regions and counting the cells expressing different combinations of colors in the source region. By performing experiments for a subset of different combinations of target regions, the counts of cells with different targeting patterns provide constraints for a system of equations that include unknown variables corresponding to the count of neurons for each projection pattern.

Thank you very much for thoroughly reading and reviewing our manuscript.

The authors propose a brilliant mathematical approach as a solution to a hard experimental problem. I express my admiration for this endeavor, and I hold the belief that this methodology will be of immeasurable value to the scientific community. The authors have effectively addressed the limitations of the study and presented compelling evidence in support of the proposed methodology, both in theoretical and experimental contexts.

Thank you for this highly positive comment recognizing the significance of our approach and its value to the scientific community. We appreciate the in-depth review of our methodology and the analyses.

Below are some suggestions to enhance the clarity and conciseness of the text in order to effectively communicate the important findings of the study to a larger audience.

Major suggestions

1. Missing use case “zero projection”

I have observed that while the authors correctly calculate all possible combinations, they have not accounted for the scenario in which cells do not project to any of the chosen target regions. It would be advisable for the authors to include this use case since some cells in a region may not project to any of the target regions under examination, and this should heavily impact the results of the analytical computations. In the event that this is not feasible biologically, the authors should provide a comprehensive explanation and justification, which I did not find in the text.

Thank you for this astute observation. Our approach indeed excludes “zero projection” types. The scope of our method is limited to the projection patterns from a source region to the *given* target regions, where the injection experiments are performed. It is true that a source region could project to a different region that is not included in the injection experiments, and the neurons that contribute to this projection could be considered as “zero projection” to the regions that *were* included in the injection experiments. However, we would like to clarify that our method aims to quantify the distinct types of projections that actually exist between a source region and a given set of target regions, and estimating the zero projection types are beyond the scope of our proposed methodology.

We recognize, as the reviewer correctly pointed out, this fact was not explicitly clarified in our original manuscript. We have revised our text (Section – 2, paragraph – 6 and Section – 4, paragraph – 2) to explain this point.

2. Visualizations of the methods and results

In order to enhance the clarity of the methodology and facilitate reader comprehension, the visualizations of both the methods and results should be improved significantly. I provide a few examples of potential improvements, but the authors have the discretion to determine the specific visualizations that are most appropriate. It is my hope that these suggestions may inspire enhancements in this area.

Thank you for the feedback and we appreciate the helpful suggestions to improve the visualizations. We have revised our visualization of injection experiments and the resulting counts in Fig. 2 and Fig S1 inspired by the reviewer's suggestion.

Figure #2:

The injections and the resulting targets are visualized with the same three colors but it is my impression that color "blue" for instance corresponds to different targets at each experiment. This is confusing as it is not clear which region is studied in each injection experiment, and the results for the same colors do not correspond to the same injected regions. For example 1944 blue in experiment one and 502 blue in experiment two do not correspond to the same region. It might be beneficial to illustrate the results for the combinations like in panel Fig2a, to have a more clear representation of the findings.

We acknowledge that our previous representations of injection experiments and counts could be difficult to comprehend for the reader. We would like to clarify that the counts 1944 and 502 denoted by blue circles in experiments 1 and 2 respectively correspond to the counts of neurons in the source region expressing blue. As the reviewer correctly pointed out, blue is injected in different regions in these experiments (region 3 in experiment 1 and region 4 in experiment 2). This means that the set of neurons (in the source region) expressing blue from experiment 1 and the set of neurons expressing blue from experiment 2 are expected to be different. We believe that our revised representations convey this better.

The visualization of combinations presented in panel 2b can be challenging to comprehend. A more effective approach would be to use an alternative representation for the combinations. For example it would be more effective to create an adjacency matrix to present the results (see attached pdf for detailed sketches). It should be noted that the representation provided is just one of many alternatives that can effectively demonstrate the same effect.

Thanks for providing helpful suggestions. Our revised representations of injection experiments and the counts are inspired by the reviewer's suggestion to use an adjacency matrix. We believe that this revision has strengthened our presentation of the methods and results.

Minor suggestions

1. Distinction between combinations versus permutations

While both refer to grouping elements of a set into subsets, in a permutation the elements of a subset are ordered, while in a combination that is not the case (all orderings are equivalent). I believe that the proposed methodology focuses on the latter and I think it would be important to clarify this distinction: the ordering of parsing of regions is not taken into account and only the presence or absence of connections between source and target regions can be investigated. This presents an opportunity for future directions to enhance the methodology, with the goal of determining the ordering of targets.

Thank you for the suggestion. It is true that our methodology only considers combinations of regions and does not take into account their permutations. The current analysis framework was developed to specifically take advantage of the combinatorial experimental design as illustrated in the manuscript. The authors will explore if there are ways to enhance the methodology utilizing a permutation design.

We have added the following sentence to clarify this distinction in Section – 2, paragraph – 3:

It is worth mentioning here that the ordering of the subset of target regions (permutations) selected in an experiment is not considered, and the proposed methodology only considers their distinct combinations.

2. Simple validation of the methodology

Since the authors have thoroughly investigated the used case of four projection targets by collecting experimental data, there is a simple validation that can be performed to demonstrate the validity of the methodology. For a subset of region targets (3) they have performed the experiments to label all three targets and computed the cell counts for each of the 7 ($2^3 - 1$) possible target combinations. Provided that sufficient experiments are available for each combination, the authors could perform the same analysis that they did for (n=4, k=3) to the (n=3, k=2) use case in which the solution could be inferred from the existing experiments. The solution for (n=3, k=2) can be analytically computed and compared to the experimental results, thus providing a trivial validation of the method and strengthening the authors' arguments.

Thank you for the suggestion. This is a valuable idea; however, we have not performed sufficient injection experiments to robustly estimate the counts of projection patterns for a 3-choose-2 model. Please note that a 3-choose-2 model with all combinations of unique experiments requires estimation of 13 unknown variables using only 9 constraints (please see the equations below). Our experimental data included in this manuscript provides a total of 6 injection experiments for the 3-choose-2 model. This requires estimation of 19 unknown variables from only 18 constraints. Thus, a robust estimation of projection pattern counts using a 3-choose-2 model is not possible using our data. On the other hand, the analysis included in the manuscript using 4-choose-3 models provided a higher constraint-to-unknown ratio (a total of 49 constraints to estimate 36 unknown variables) for more robust results.

We are including here the results from our analysis of data using the 3-choose-2 model (Table – r1) for review. The constraints included counts from injection experiments performed in regions MOp (1), MOs (2), and SSp (3). We would like to point out that the trend of solution convergence (medians) indeed suggests, at least for some projection patterns (T1, T3 and T13), a convergence towards the solutions we obtained for the 4-choose-3 system. However, the IQR is high for many projection patterns as expected due to the under-constrained nature of the 3-choose-2 model. Therefore, these results are not included in the revised manuscript to validate the results that were previously obtained using the 4-choose-3 model.

Equations for 3C2 model:

$$\text{inj_12_comb_1} = ((T2 + T23) *(a2)) + ((T12 + T123) *(a2 * (1-a1)))$$

$$\text{inj_12_comb_2} = ((T1 + T13) *(a1)) + ((T12 + T123) *(a1 * (1-a2)))$$

$$\text{inj_12_comb_3} = ((T12 + T123) *(a1 * a2))$$

$$\text{inj_13_comb_1} = ((T3 + T23) *(b3))+((T13 + T123) *(b3 * (1-b1)))$$

$$\text{inj_13_comb_2} = ((T1 + T12) *(b1))+((T13 + T123) *(b1 * (1-b3)))$$

$$\text{inj_13_comb_3} = ((T13 + T123) *(b1 * b3))$$

$$\text{inj_23_comb_1} = ((T3 + T13) *(c3))+((T23 + T123) *(c3 * (1-c2)))$$

$$\text{inj_23_comb_2} = ((T2 + T12) *(c2))+((T23 + T123) *(c2 * (1-c3)))$$

$$\text{inj_23_comb_3} = ((T23 + T123) *(c2 * c3))$$

Table – r1. Estimated counts of projection patterns using a 3-choose-2 model

Projection Pattern, Median, IQR

T1, 1391, 201

T2, 2226, 369

T12, 112, 118

T3, 7041, 3312

T13, 649, 345

T23, 85, 58

T123, 58, 19

SUM, 11689, 3844

3. Distinction between experimental and theoretical results

To prevent confusion between experimental and computational results, it would be beneficial for the reader if the authors clearly distinguished between the two in both the text and figures. This will enable the reader to discern the type of results being presented with ease.

Thank you for this valuable comment. We have made the distinction between experimental and computational results at several places in the revised manuscript. This includes lines 147, 166, 185, 206 and 216 in the text, in figure 2 and S1 captions, and in the table – 1 title.

I would like to commend the authors for their meticulous work and dedication to this manuscript, and express my best wishes for their continued progress in this direction of valuable research.

Lida Kanari

Thank you very much for this encouraging comment. The authors are grateful for your strong interest and dedication in reviewing this manuscript and for providing us with several highly valuable suggestions to improve the quality of this manuscript.

Reviewer #3 (Remarks to the Author):

The authors provide a compelling analytical framework to determine the distinct projection profiles of single neurons from one source region to a set of target regions. This is potentially a very powerful and useful development. I do not have the background to determine the mathematical validity of the approach, but it appears well reasoned, developed and I see no obvious flaws. However, I have two major concerns about the work regarding its suitability to the wide-readership of Nature Communications.

Thank you for the review of and positive comments on our manuscript. We address the concerns below and in the revised resubmission.

First, the paper does not provide a ground truth or orthogonal data set to demonstrate that a combination of practical, even if difficult, experiments and the proposed analytical approach will capture the different projection patterns of individual neurons. Both single cell reconstructions and MAPseq could potentially provide these datasets.

We thank the Reviewer for this constructive comment. The authors clearly appreciate the advancement of single neuron reconstructions and MAPseq. In fact, one recent influential Nature publication by the BICCN anatomy working group (doi: 10.1038/s41586-021-03970-w) include connectivity data generated with classic anterograde tracing, cell type-specific viral tracing, multi-fluorescent retrograde tracing, single neuron reconstruction and MAPseq methods. This joint anatomy paper demonstrated that these multi-modality data provide a full spectrum of cell type-specific connectome for a given brain structure, e.g., the MOp. Both Drs. Ascoli and Dong, the senior authors of this submission, were also the co-corresponding authors for that paper. Although we greatly value the complementary nature of these technologies, we must note that that effort involved a truly exceptional coordination effort by a team of 80 coauthors from over a dozen independent laboratories.

In theory, single cell reconstructions could provide a ground truth for classifying neuron types and map their projectome (as suggested by the Reviewer). In reality, however, it might not always be practical. Currently, most available single neuron morphology data in the field come from two major sources: the Janelia MouseLight and BICCN fMOST projects. To generate these data, single neurons were labeled using AAV-based sparse labeling methods in different Cre-driver mouse lines, e.g., Cux2-Cre for layer 2/3 IT neurons, or Plxind1-Cre for layer 5 neurons. For a given brain structure, e.g., the MOp, the sampling of reconstructed neurons or cell types heavily rely on which Cre lines are used. Moreover, most Cre lines may be good genetic markers but might not represent comprehensive cell types. Other facts can also affect the quality of reconstructed neurons, such as the volume and titer of virus, imaging quality, reconstruction algorithm, and manual validation. Thus, it still remains challenging to collect a sufficient number of reconstructed neurons with good quality that comprehensively represent all projection neuron types for a given brain structure, e.g., MOp. Notably, there were only ~300 reconstructed neurons in the above-referenced joint BICCN anatomy paper, and to the best of our knowledge, this still constitutes the extent of the existing dataset available for this brain area (now referred to at lines 264-273). Considering also possible differences in brain registration at the sub-regional level, we find it impossible to reliably ascertain a significant overlap between the specific MOp area in which the somata of the BICCN reconstructions are located and the multi-labeled cell bodies obtained in the experiments described in our present Combinatorial Projectomics paper. Therefore, the only way to satisfy the request to validate our results with single-cell reconstructions would be to perform a dedicated full set

of new injections and single-cell reconstructions from the locations of our detected somata. This would constitute a completely different large-scale project that we are not equipped to embark upon.

MAPseq (and Barseq), developed by the laboratory of Dr. Tony Zador, has the great advantage of mapping the cell type-specific projectome at high throughput as demonstrated in their own papers and our BICCN joint anatomy paper. At the same time, our assessment of the existing MapSeq data from our joint BICCN anatomy paper is again that there is not high confidence that the specific MOp sub-area in which the somata of the BICCN bar-coded neurons are located significantly overlaps with that of the multi-labeled cell bodies obtained in the experiments described in our Combinatorial Projectomics paper. Thus, the only way to satisfy the request to validate our results with MapSeq would be to perform a dedicated set of new anterograde injections from the locations of our detected somata with subsequent laser-captured physical dissection and bar-code sequencing from the target areas. Furthermore, MAPseq also has limitations: MAPseq data collection relies on previous knowledge or anterograde tracing data as guidance to dissect the targeted brain regions for acquiring transported barcoding molecules. Thus, the granularity and accuracy of MAPseq-based projectome is heavily dependent on tissue dissection sampling densities and anatomical accuracy of each targeted area. Because tissue dissections are conducted in fresh frozen, thick sections without histological staining (e.g., Nissl staining), it remains very challenging to dissect tissues accurately following their anatomical borders (especially for those smaller brain structures). For example, it is possible that the dissected “SSp” tissue might include a small piece of adjacent “MOp” tissue, which will cause misinterpretation of data. For the above reasons, we do not feel that MAPseq data can be used as a “ground truth”. We added a dedicated paragraph in the Discussion to address the above concern (lines 277-290).

Despite our deep understanding of those technological limitations, Drs. Ascoli and Dong are both strong advocates for applying single neuron reconstruction and MAPseq to improve the throughput, scalability, and accuracy of constructing a cell type-specific connectome or projectome of the mouse brain. Our Nature 2021 BICCN joint anatomy paper shows how these technologies can be applied together for constructing connectomes/projectomes, since there is no “all-in-one” or “perfect” methods that can achieve this daunting task. The new approach reported in this current manuscript is another independently valuable effort towards this same direction. Together with the broad BICCN community, we hope to apply Combinatorial Projectomics along with all of the available toolsets of anterograde tracing, viral labeling, single neuron reconstruction, and MAPseq, to systematically characterize cell type-specific projectomes for different brain structures. Based on our experience, these comprehensive multimodal approaches require a much larger effort that goes clearly beyond the scope of this collaborative study between a single computational lab and a single experimental lab.

Second, the authors acknowledge that producing data sets with more than 3-4 colors is about the limit of what is possible. Given that many brain areas, including the one investigated usually innervate at least 10, and can innervate more than 20 targets, it remains unclear how widely applicable or scalable this approach currently is, or will be in the future.

We thank the reviewer for the constructive comments about the feasibility and robustness of our approach and also appreciate the valuable concern about its scalability and accuracy/precision. Retrograde tract tracing methods have been broadly used in the neuroscience field for nearly 4 decades (since early 80s). The Dong lab is one of the pioneer groups to systematically apply multiple fluorescent retrograde tracers, mostly 3-4 colors, in constructing the mouse connectome (e.g., Zingg, et al., 2014;

Bienkowski, et al., 2018; and BICCN joint anatomy paper). Through these publications, we have demonstrated the feasibility, robustness, and accuracy/precision of this multi-fluorescent retrograde tract tracing method, in parallel to anterograde tracing and other methods. In terms of scalability, the Dong lab has conducted triple or quadruple fluorescent retrograde tract tracing experiments in over ~600 mice; those injection sites are spread across many brain structures and were collected to validate anterograde tract tracing results. A large collection of these data has been deposited in, and is publicly available from, the Brain Image Library (BIL). The approach presented in this paper is designed for a different purpose: revealing the cell type-specific projectome via well-established retrograde tracing techniques. We do not anticipate any difficulties to scale-up. For example, based on our own past and ongoing experimental work, this technique can be scaled to perform 500 quadruple injection experiments in a reliable way, assuming the resources of a large effort such as those presently pursued by the Allen Institute, Janelia, and NIH BICCN. Doing so could resolve up to 12 distinct targets, potentially quantifying up to >4000 different projection types at once in each and every source region projecting to those targets. We added a dedicated paragraph in the Discussion to address the above concern (lines 306-320).

I believe the authors have demonstrated the feasibility and robustness of their approach, but I do not believe they have demonstrated how it scales or its accuracy/precision. The scaling problem is mainly experimental that can either involve being able to perform both many accurate injections and subsequently detect and separate the many colors in the source brain region; or performing many sets of 3-4 color experiments to reveal the full set of projection patterns. To demonstrate the accuracy and precision of the method requires other data sets, e.g. single cell reconstructions and/or barcoding methods such as MAPseq. The authors eloquently address the difficulty of this issue in the discussion, highlighting the need for new approaches. However, I do feel it needs to be addressed more directly and thoroughly than is presented in the manuscript.

We should note here that the primary scope of the paper is analytical and not experimental. As this reviewer acknowledges, we have introduced a powerful new method and demonstrated computationally the robustness of our approach. In order to show that the methodology is practically applicable to a real-world empirical setting, we have also performed a full set of experiments paralleling the numerical analysis of surrogate data. As the reviewer wrote, the results demonstrate feasibility because the analysis, as theorized, proves to work in practice on anatomical measurements affected by all sources of error. This completely fulfills the analytical scope of the paper. Because the primary scope of the paper was not to produce experimental data specifically regarding the analyzed neurons, it seems beyond reasonability to multiply the empirical validation effort, which would also distort the primary focus of this innovation. Nevertheless, we point out in the Discussion that our experimental results are indeed consistent with those reported in the literature (lines 258-264).

This work is clearly of high quality but as currently presented appears more suited to a more specialized journal.

We are gratified for the assessment of high quality, and we hope that the Editor will agree that the improved Discussion in the revised resubmission frames the Results in a useful context for the broad readership of Nature Communications. As part of a special Nature package to highlight the neuroanatomy advances of the major NIH-funded BICCN effort, we believe that the unique, yet synergistic, contribution of this paper is especially valuable for the scientific community.

REVIEWERS' COMMENTS

Reviewer #1 (Remarks to the Author):

Reviewer 1 accept the responses and looking forward to see this paper published incorporated the more detailed suggestions by reviewers 2 and 3.

Reviewer #2 (Remarks to the Author):

I congratulate the authors for the thorough revision of their manuscript. The latest version exhibits enhanced readability and effectively conveys the important message of their work with utmost clarity. The figures have undergone substantial improvements, rendering them significantly more comprehensible. Based on the merits of its current form, I wholeheartedly recommend the publication of this manuscript. Nonetheless, I would like to suggest a few minor points that could be further improved in this version.

Minor comments:

1. Figure 2: It is not clear what the two numbers indicate in panel B for each experiment. It would be helpful to explain in more detail in the caption what the two numbers indicate.
2. The statistical measurement of IQR is not clearly explained in the methodology and therefore it is not clear what it refers to. It would be useful to explain this measurement in the methodology for clarity.
3. Definition of the error in methodology is not fully explained. It is not clear how the true and estimated counts for each type are computed for each experiment, in order to quantify the error that is used as a measure of validation for the methodology.

Reviewer #3 (Remarks to the Author):

The authors have taken all reviewers comments into consideration and have improved the manuscript. This work remains of high quality and the new figures and discussion improve its clarity and make clear where the short comings lie.

While I have no scientific objections to this work being published

I remain concerned that the scalability of the experiments needed to make use of the proposed analytical approach is not demonstrated directly. I believe this omission limits the papers potential impact and interest for the broad readership of Nature Communications.

Reviewer #2 (Remarks to the Author):

I congratulate the authors for the thorough revision of their manuscript. The latest version exhibits enhanced readability and effectively conveys the important message of their work with utmost clarity. The figures have undergone substantial improvements, rendering them significantly more comprehensible. Based on the merits of its current form, I wholeheartedly recommend the publication of this manuscript. Nonetheless, I would like to suggest a few minor points that could be further improved in this version.

The authors appreciate your interest in our work and the in-depth review of our manuscript, which improved its quality. We have further revised our manuscript based on your minor comments.

Minor comments:

1. Figure 2: It is not clear what the two numbers indicate in panel B for each experiment. It would be helpful to explain in more detail in the caption what the two numbers indicate.

Thanks for the comment. We have added 2 sentences to explain this in more detail in figure 2 caption:

“Figure 2. Simulations of triple injection experiments for computational analysis. a. Surrogate population sizes for 15 projection patterns. b. 4 distinct combinations of triple injections and repeated trials of each were simulated using values sampled from a normal distribution with mean 0.6 and standard deviation 0.2. **These values correspond to the fractions of projecting neurons that are in fact labeled.** Two entries in each cell correspond to the counts from two trials of the same experiment for the population sizes given in panel a. **Note that the different counts are due to the differences in the fractions of labeled neurons between two trials of the same experiment.** “

2. The statistical measurement of IQR is not clearly explained in the methodology and therefore it is not clear what it refers to. It would be useful to explain this measurement in the methodology for clarity.

Thanks for the comment. We have added a brief explanation in paragraph-3 of section “Solving the systems of equation” in Methods.

“...In addition to the medians, interquartile ranges (IQR) were calculated from the top 10 EA solutions for the analysis involving experimental data. The IQR characterized the spread of estimated solutions in the parameter space for each projection pattern.”

3. Definition of the error in methodology is not fully explained. It is not clear how the true and estimated counts for each type are computed for each experiment, in order to quantify the error that is used as a measure of validation for the methodology.

Thanks for the comment. We have expanded paragraphs 2 and 3 of section “Solving the systems of equation” in Methods to fully explain all the errors utilized in this study. The revised section further provides more clarity on how the true and estimated counts are computed for each experiment.